# Effect of Maternal Retinol Status at Time of Term Delivery on Retinol Placental Concentration, Intrauterine Transfer Rate, and Newborn Retinol Status

**DOI:** 10.3390/biomedicines8090321

**Published:** 2020-08-31

**Authors:** Melissa Thoene, Haley Haskett, Jeremy Furtado, Maranda Thompson, Matthew Van Ormer, Corrine Hanson, Ann Anderson-Berry

**Affiliations:** 1Department of Pediatrics, University of Nebraska Medical Center, 981205 Nebraska Medical Center, Omaha, NE 68198-1205, USA; haley.haskett@gmail.com (H.H.); maranda.thompson@unmc.edu (M.T.); matthew.vanormer@unmc.edu (M.V.O.); alanders@unmc.edu (A.A.-B.); 2Department of Nutrition, Harvard T. Chan School of Public Health, 655 Huntington Avenue, Boston, MA 02215, USA; jfurtado@hsph.harvard.edu; 3College of Allied Health Professions, University of Nebraska Medical Center, 984045 Nebraska Medical Center, Omaha, NE 68198-4045, USA; ckhanson@unmc.edu

**Keywords:** intrauterine, maternal, infant, newborn, placenta, deficient

## Abstract

Retinol (vitamin A) is essential, so the objective of this Institutional Review Board approved study is to evaluate retinol placental concentration, intrauterine transfer, and neonatal status at time of term delivery between cases of maternal retinol adequacy, insufficiency, and deficiency in a United States population. Birth information and biological samples were collected for mother–infant dyads (*n* = 260). Maternal and umbilical cord blood retinol concentrations (*n* = 260) were analyzed by HPLC and categorized: deficient (≤0.7 umol/L), insufficient (>0.7–1.05 umol/L), adequate (>1.05 umol/L). Intrauterine transfer rate was calculated: (umbilical cord blood retinol concentration/maternal retinol concentration) × 100. Non-parametric statistics used include Spearman’s correlations, Mann–Whitney U, and Kruskal–Wallis tests. *p*-values <0.05 were statistically significant. Only 51.2% of mothers were retinol adequate, with 38.4% insufficient, 10.4% deficient. Only 1.5% of infants were retinol adequate. Placental concentrations (*n* = 73) differed between adequate vs. deficient mothers (median 0.13 vs. 0.10 μg/g; *p* = 0.003). Umbilical cord blood concentrations were similar between deficient, insufficient, and adequate mothers (0.61 vs. 0.55 vs. 0.57 μmol/L; *p* = 0.35). Intrauterine transfer increased with maternal deficiency (103.4%) and insufficiency (61.2%) compared to adequacy (43.1%), *p* < 0.0001. Results indicate that intrauterine transfer rate is augmented in cases of maternal retinol inadequacy, leading to similar concentrations in umbilical cord blood at term delivery.

## 1. Introduction

Retinol (vitamin A) is an essential nutrient throughout the life cycle [1,2]. However, its influence is particularly critical during periods when cells are rapidly proliferating and differentiating, such as during pregnancy. Adequate maternal retinol stores are imperative for fetal growth and maturation of multiple organ systems, including the immune system, lungs, eyes, and thyroid [2,3,4]. Maternal retinol deficiency increases the risk of complications during pregnancy and the postpartum period and has been positively associated with maternal infections, night blindness, anemia, and birth defects [2,5,6]. Maternal retinol deficiency also has significant consequences for the newborn including impaired immune function and increased morbidity and mortality due to neonatal infections, congenital diaphragmatic hernia, and bronchopulmonary dysplasia [2,7,8,9]. The recommended dietary allowance of vitamin A (as measured by retinol activity equivalents, or RAE) in 19–50-year-old women increases from 700 to 770 micrograms (μg)/day during pregnancy to meet increased demands for fetal growth [1]. Due to the increased need during pregnancy, maternal stores may be progressively depleted if dietary intake is persistently inadequate.

Retinol deficiency is one of the most significant nutritional problems and has been recognized for decades as a public health concern in developing countries [10]. Annually and worldwide, it affects 19 million pregnant women and contributes to early death in 650,000 children [11,12]. Due to high gross domestic product, there has been limited assessment of retinol status in the United States (U.S.) population, as assumptions are that the country remains fairly exempt from deficiency [11]. However, a recent study assessing U.S. pregnant women at time of delivery (*n* = 189) identified that 41% were insufficient and 10% were deficient [13], according to criteria set by the World Health Organization (WHO) [11].

The rapidly developing fetus depends on the placenta for intrauterine transfer of circulating maternal retinol. The transfer is tightly regulated, as inadequate or superfluous substrate may be harmful to the fetus, with special consideration for teratogenic consequences resulting from excess retinol [2]. Prior research describes the presence of retinol receptors on the brush border membrane of the placenta, implying the placenta may have regulatory mechanisms for transfer [14,15,16]. In a clinical example, research such as that by Dimenstein et al. reports similar retinol concentrations in placenta and umbilical cord blood between women (*n* = 31) with adequate vs. subadequate retinol status at time of delivery [15]. However, maternal to fetal retinol transfer has not been evaluated in a developed country, nor has it been compared between deficient vs. insufficient retinol statuses as defined by the WHO [11]. Therefore, the objective of this study is to evaluate retinol placental concentration, intrauterine transfer rate, and neonatal status at the time of term delivery between cases of maternal retinol adequacy, insufficiency, and deficiency in a U.S. population.

## 2. Materials and Methods

### 2.1. Study Design and Participants

This study was approved by the Institutional Review Board at the University of Nebraska Medical Center (Omaha, NE, IRB protocol# 112-15-EP, approved on 14 April 2015and conducted in accordance with the Declaration of Helsinki. Mother–infant dyads were screened and approached for informed written consent for study enrollment at time of delivery between 2015 and 2017. Mothers provided consent for both themselves and their infants. Only mothers delivering at term (≥37 weeks gestational age [17]) were included in this study as gestational age may impact nutrient delivery due to increasing maternal blood volume and placental surface area [18]. Inclusion criteria included mothers at least 19 years of age, delivering at least one live-born infant, and able to make their own medical decisions. Mothers and infants must be free of congenital abnormalities, an inborn error of metabolism, or a gastrointestinal, liver, or kidney disease that may impair normal nutrient metabolism. Infants of multiple gestation were eligible for inclusion, however, only information was included for the first-born twin. No infants deemed ward of state were approached for consent. No incentives were offered so as not to influence the decision to participate.

### 2.2. Biological Samples Collection

Only dyads with both maternal serum and umbilical cord blood were included for this analysis. Retinol concentrations were analyzed from samples of placenta (as available; target volume 10 g), maternal blood (target 5 mL), and umbilical cord blood (target 1 mL) taken at time of delivery for enrolled mother–infant dyads. Both maternal blood and umbilical cord blood is collected as standard of care during hospitalization for delivery, so samples of each were collected for study purposes. Collected study samples were protected from light with use of an amber-colored bag. They were processed immediately within 12 h of collection and stored at –80 °C.

### 2.3. Evaluation of Retinol Concentration

The Nutritional Biomarker Laboratory at the Harvard T.H. Chan School of Public Health analyzed retinol concentrations using high-performance liquid chromatography (HPLC) per methods described by El-Sohemy et al., but with some modifications [19]. Samples of placenta tissue were weighed, then homogenized in distilled, deionized water by mechanical pulverization (Polytron PT1200, Kinematica AG, Lucerne, Switzerland) to form an aqueous slurry. The samples of placenta and plasma serum were mixed with ethanol containing 10 mg/mL rac-Tocopherol (Tocol) as an internal standard, extracted with hexane, evaporated to dryness under nitrogen, and redissolved in ethanol, dioxane, and acetonitrile. Samples were analyzed by HPLC using a Restek Ultra C18 150 mm × 4.6 mm column, 3 μm particle size (Restek, Corp. Bellefonte, PA, USA) encased in a Hitachi L-2350 column oven (Hitachi, San Jose, CA, USA). This prevents temperature fluctuations and is equipped with a trident guard cartridge system. A mixture of acetonitrile, tetrahydrofuran, methanol, and a 1% ammonium acetate solution (68:22:7:3) were used as mobile phase at a flow rate of 1.1 mL/min, using a Hitachi Elite LaChrom HPLC system comprised of an L-2130 pump in isocratic mode, an L-2455 Diode Array Detector (monitoring at 300 nm and 445 nm), and a programmable AS-2200 auto-sampler with a chilled sample tray. The system manager software (D-7000, Version 3.0; Hitachi, San Jose, CA, USA) was used for peak integration and data acquisition. The minimum detection limits (MDLs) in serum are for 5.23 for retinol (μg/L). Internal quality control was monitored with four control samples analyzed with each run. External quality control was monitored by participation in the standardization program for retinol analysis from the National Institute of Standards and Technology, Unites States of America. Placenta retinol concentrations were reported in microgram/gram (μg/g) and blood retinol concentrations were reported in micromole/Liter (μg/L).

Based on blood concentrations, categories of retinol status for this analysis were defined as follows: deficient (≤0.70 μmol/L), insufficient (>0.70–1.05 μmol/L), and adequate (>1.05 μmol/L). These value categories are similar to those set by the World Health Organization [11] and previous research assessing retinol status in a U.S. perinatal population [13]. Originally, initial analysis of maternal serum and umbilical cord blood retinol concentrations was reported in μg/L, but was subsequently converted to μmol/L by the following calculation: ((μg/L)/(286.45 g/mol [20])). Per our terminology, intrauterine transfer rate is the percent of infant nutrient concentration compared to maternal concentration, which indicates the rate of transplacental nutrient transfer. Therefore, intrauterine retinol transfer rate was calculated as follows: ((umbilical cord blood retinol concentration μg/L)/(maternal retinol concentration μg/L)) × 100.

### 2.4. Birth and Demographic Data Collection

Birth and demographic data were collected for mother–infant dyads at time of delivery from the electronic health system. Maternal data collected included age (years), race, and mode of delivery (vaginal vs. Cesarean). Infant data included gender, birth gestational age (reported in weeks and days), and birth weight (grams, g). Mothers also completed the validated Harvard Willett food frequency questionnaire at time of hospitalization to evaluate dietary intake and supplement use over the past one year [21]. Results provided data regarding the usual maternal dietary intake of retinol, as measured in RAE (μg/day).

### 2.5. Statistical Analysis

Due to skewed histograms for various biological retinol samples and intrauterine transfer rates, non-parametric tests were utilized. Non-parametric tests were also utilized for remaining data to maintain consistency in reporting. Descriptive data are reported in counts, proportions, median, minimum, maximum, and interquartile range (IQR). Spearman’s correlation coefficients assessed relationships between two continuous variables. The Kruskal–Wallis test compared variance of continuous data between three or more categorical groups and the Mann–Whitney U test compared continuous values between two groups. IBM SPSS Statistics for Windows, Version 26.0 software (IBM Corp., New York, NY, USA) was used for all statistical analysis. A *p*-value of <0.05 was considered statistically significant.

## 3. Results

### 3.1. Demographic & Birth Characteristic Data

There were 260 mother–infant dyads included in this analysis. Among these, 73 placental samples were available for analysis. Continuous response data for mother–infant pairs are displayed in Table 1. Categorical response data and frequencies are displayed in Table 2.

### 3.2. Blood and Placental Retinol Results

Overall, the median maternal serum retinol concentration was 1.07 μmol/L (range 0.30–2.13) with median umbilical cord serum concentration at 0.57 μmol/L (range 0.05–1.29) and intrauterine retinol transfer rate 52.6% (range 4.3%–430.0%). Median placental retinol concentration was 0.13 μg/g (range 0.04–0.82). Maternal serum retinol correlated with placental retinol (*r* = 0.38, *p* = 0.001) but not with umbilical cord blood retinol (*p* = 0.33). There was no correlation between retinol concentrations in placenta and umbilical cord blood (*p* = 0.78). There were no significant correlations between retinol concentrations in maternal serum (*p* = 0.77), umbilical cord blood (*p* = 0.50), or placenta (*p* = 0.75) with maternal dietary RAE intake. Table 3 displays the prevalence of retinol deficiency, insufficiency, and adequacy in mothers and infants. Table 4 displays the proportions of infant retinol status based on maternal retinol status.

### 3.3. Comparisons between Maternal Retinol Categories

#### 3.3.1. Retinol Concentrations and Intrauterine Transfer

Comparison of blood and placental retinol concentrations and intrauterine transfer rates between maternal retinol categories are listed in Table 5. Blood retinol concentrations of maternal serum and umbilical cord blood according to maternal retinol category are represented in Figure 1. Among the available placental samples, 11 were from retinol deficient mothers, 25 insufficient, and 37 adequate.

#### 3.3.2. Maternal Dietary Intake

Maternal RAE dietary intake did not differ between maternal retinol categories (median 1422 vs. 1566 vs. 1329 μg/day, for deficient, insufficient, and adequate categories, respectively; *p* = 0.25).

#### 3.3.3. Gestational Age at Birth

There were no differences in gestational age at birth between maternal categories of retinol deficiency, insufficiency, or adequacy (median 39 1/7 vs. 39 5/7 vs. 39 5/7 weeks; *p* = 0.07).

## 4. Discussion

### 4.1. Intrauterine Retinol Transfer

Similar to previous research in a developing country [15], study results within this U.S. population demonstrate that intrauterine retinol transfer is increased in cases of maternal retinol inadequacy. Novel, however, are study results demonstrating that intrauterine transfer is further augmented when mothers are retinol deficient compared to insufficient. Remarkably from results, raw median levels of retinol in umbilical cord blood were highest among infants born to retinol deficient (0.61 umol/L) compared to insufficient or adequate mothers (0.55 and 0.57 umol/L). It is unknown if this is an added protective effect for infants born to mothers with true retinol deficiency compared to insufficiency or adequacy. Ultimately, however, newborn retinol concentrations remained similar across varying statuses of maternal retinol given there was no statistical differences between groups. Thus, results primarily indicate that intrauterine retinol transfer from mother to infant is highly regulated, but at the expense of maternal stores. While maternal retinol levels at the start of pregnancy are unknown for this cohort, it is possible that maternal levels declined during pregnancy to meet fetal nutrient needs, even decreasing from adequate to insufficient or insufficient to deficient reference ranges.

### 4.2. Retinol Status at Delivery/Birth

Results from this maternal population endorse that a “moderate” public health problem related to vitamin A deficiency (≥10% to <20% of the population) exists within a developed country [11]. Comparatively, prevalence of vitamin A deficiency in this U.S. population is significantly higher than the 2.0% estimated between 1995 and 2005 (defined as serum retinol <0.70 µmol/L) [11].

Proportions of infant retinol deficiency, insufficiency, and adequacy showed similar distributions across maternal retinol categories. However, only 1.5% of these term-born infants met adult criteria for retinol adequacy. While the WHO acknowledges that newborns begin life with low retinol concentrations [22], there remains no well-defined reference range for true deficiency in a newborn population. Per the WHO, vitamin A deficiency in children aged 6–59 months is identified as a “problem” when ≥20% meet criteria (defined as serum retinol concentration ≤0.70 µmol/L) [23]. Though 75.0% of infants in this study met deficiency criteria, the age of this newborn population limits applicability. Study results consequently pose the question of if neonatal retinol concentrations can be compared to reference ranges for older-aged populations to accurately identify retinol deficiency, insufficiency, or adequacy, which has been previously debated [24]. Given that only 1.5% of this study infant population met the adequacy criteria for older-aged populations, it may be inquired if low (term-born) newborn concentrations do not present similar adverse effects as those seen in older populations, indicating lower reference ranges may be tolerable in early life. In further evidence from this study, among mothers with retinol concentrations one standard deviation above the adequate concentration cutoff (>1.4 µmol/L), only 2.6% of their infants (*n* = 1/39) met criteria for adequacy, with 64.1% (*n* = 25/39) classified as deficient. Despite having lower serum retinol levels compared to adult populations, the WHO does not currently recommend vitamin A supplementation for infants aged 1–5 months, stating supplementation does not provide a benefit in reducing illness or death [22].

Nonetheless, newborn retinol deficiency or potential risks therein cannot be understated. Among study infants, 5.8% (*n* = 15/260) met criteria for severe deficiency based on older-aged reference ranges (retinol concentration ≤0.35 µmol/L) [11]. Of these infants, 73.3% (*n* = 11/15) were born to retinol adequate mothers, followed by 20.0% (*n* = 3/15) to insufficient mothers, and 6.7% (*n* = 1/15) to deficient mothers. Therefore, results indicate that modulation of intrauterine nutrient transfer may not prevent against all potential neonatal nutrient deficiencies and also highlights the importance of adequate nutrition for mother–infant dyads from all time points between preconception to postpartum. These findings may further suggest that there are additional, and potentially unidentified, determinants of intrauterine retinol transfer that influence newborn retinol status.

While placental retinol concentrations showed no relationship with levels in umbilical cord blood, they were moderately correlated with those in maternal serum. Placental retinol concentrations were not significantly different between retinol deficient vs. insufficient mothers. However, median placental retinol concentrations were 1.3 times higher in retinol adequate vs. retinol deficient mothers. Comparatively, Diminstein et al. reported no significant differences in placental concentrations between maternal retinol adequate vs. subadequate groups [15]. While there is no significant correlation between retinol in placenta and umbilical cord blood in this study, it remains unknown what effect increased placental retinol concentrations have on the organogenesis or immunity status of the placenta, which has latent consequences to the fetus. Mechanisms of transplacental retinol transfer and regulation are not fully understood. However, further research is necessary to understand how maternal deficiency at varying stages of pregnancy may impact placental size and development, given retinol supports gene expression and the integrity of epithelial cells—especially during a time of rapid proliferation [2]. If research identifies maternal retinol deficiency to hinder proper placental development, fetal growth must also be assessed as altered placental development could impair appropriate oxygen and nutrient exchange.

### 4.3. Retinol Dietary Intake

In this cohort, 14.0% (*n* = 31/221) of mothers who completed the food frequency questionnaire did not meet the recommended RAE intakes during pregnancy of 770 μg/day [1]. Of interest, maternal dietary intake of RAE did not correlate with maternal serum retinol concentrations and dietary intake was not significantly different between maternal retinol categories. However, there may be rationale for these findings given that approximately 90% of retinol is stored in the liver when dietary intake is adequate [2]. Liver stores support retinol homeostasis when dietary intake is chronically inadequate, so one-sample blood concentrations may not reflect current dietary intake while there is adequate liver reserve. Additional nutrients including zinc, iron, and dietary fat may also alter absorption or status of retinol [1], confounding cross-sectional analysis. Furthermore, biomarkers of infection or inflammation, such as those generated by pregnancy-induced disorders, demonstrate inverse relationships with serum retinol levels [25]. Though these biomarkers from this delivering cohort were not analyzed concurrent with presented results, they may provide reasoning for variable serum retinol levels despite similar dietary intakes. Regardless, results ultimately highlight the importance of adequate maternal retinol status prior to conception to prevent maternal deficiency during pregnancy.

Newborns rely on external sources of retinol from maternal milk or infant formula to meet essential needs following birth. Therefore, adequate dietary intake also remains important during pregnancy to prevent deficiency at the start of lactation. The recommended dietary allowance for maternal RAE intake increases from 770 to 1300 μg/day from pregnancy to lactation to support retinol transfer via human milk provision [1]. Retinol concentrations in breast milk can even be analyzed to determine retinol status on a population level, with milk levels ≤1.05 μmol/L indicating inadequacy [25]. Consequently, it is expected that retinol stores are further depleted through lactation for retinol deficient or insufficient mothers with persistently inadequate dietary intake. Similarly, mothers with lower serum retinol concentrations produce lower retinol-containing milk for their infants [26], which increases the risk for infant deficiency.

Vitamin A supplementation during pregnancy is not universally recommended due to risk of teratogenic effects of excessive intake. The WHO only recommends supplementation to prevent night blindness in areas where ≥5% of delivering women have a history of night blindness during the previous 3–5 years or if ≥20% of pregnant women meet criteria for deficiency based on serum levels [27]. This cohort yielded retinol deficiency rates at <20%, so alternative strategies may be considered to optimize retinol status beyond supplementation. In example, tolerable upper limits for dietary intake are 2800–3000 μg/day of preformed vitamin A, such as that coming from animal products or dietary supplements containing retinol or retinol esters [28]. However, many alternative foods (i.e., fruits and vegetables) contain provitamin A carotenoids (i.e., α-carotene, β-carotene, β-cryptoxanthin) that can be converted to retinol in the body [29]. Unlike preformed vitamin A, these provitamin A carotenoids have not been associated with the same risk of reproductive teratogenicity [28]. Therefore, nutrition counseling for women during pregnancy could promote higher dietary intakes of provitamin A carotenoids, seeking to optimize retinol status without increasing risk of toxicity.

### 4.4. Strengths and Limitations

The primary strength of this study is that it enhances understanding of intrauterine retinol transfer and resulting newborn retinol status among a diverse population living in a developed country. In addition to comparison of blood retinol levels between mother and infant, retinol concentrations in placenta allow for greater understanding of intrauterine nutrient transfer. Limitations include that this is a cross-sectional study, so retinol status and maternal RAE intake pre- and postpartum are unknown. Lastly, this study only included infants born at term gestational ages, so further research is needed to understand how retinol status and intrauterine transfer changes at lower gestation and placental development.

## 5. Conclusions

These results indicate that intrauterine transfer rate is augmented in cases of maternal retinol deficiency or insufficiency, leading to similar concentrations in umbilical cord blood at the time of term delivery. Further research is necessary to understand the mechanisms involved in the modulation of intrauterine transfer.

## Figures and Tables

**Figure 1 biomedicines-08-00321-f001:**
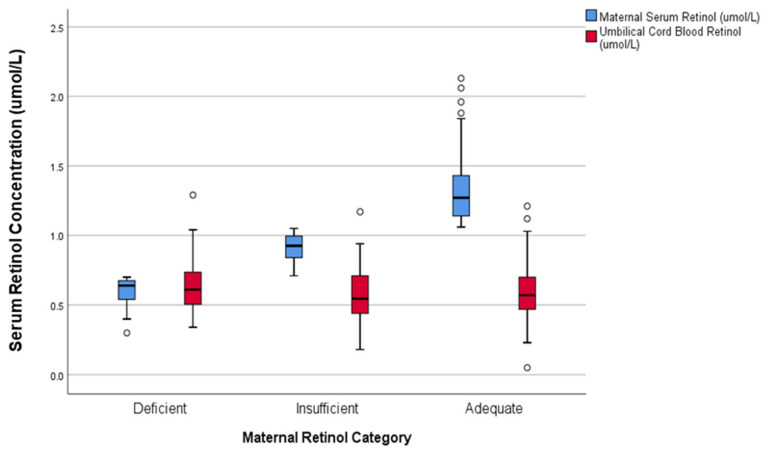
Comparison of maternal and infant retinol concentrations (micromoles/Liter) by maternal retinol status category.

**Table 1 biomedicines-08-00321-t001:** Continuous responses of maternal and infant demographic and birth characteristics.

	*n*	Median	IQR	Minimum	Maximum
**Gestational Age at Birth ^a^**	260	39 + 4	2 + 0	37 + 0	42 + 1
**Maternal Age (Years)**	259	29	8	19	44
**Maternal RAE ^b^/Day Intake (μg)**	221	1402	1098	188	4946
**Birth Weight (g)**	260	3453	575	1907	4622

^a^ Reported in “weeks + days”; ^b^ retinol activity equivalents. IQR, interquartile range.

**Table 2 biomedicines-08-00321-t002:** Categorical continuous responses of maternal and infant demographic and birth characteristics.

	*n*	Frequency (%)
**Infant Gender**	260	120 female (46.2%)
		140 male (53.8%)
**Mode of Delivery**	259	187 vaginal (72.2%)
		72 Cesarean (27.8%)
**Race**	260	170 Caucasian (65.4%)
		40 African American (15.4%)
		25 Hispanic (9.6%)
		6 Asian/Pacific Islander (2.3%)
		1 American Indian (0.4%)
		18 Other/Unknown (6.9%)

**Table 3 biomedicines-08-00321-t003:** Retinol status of mothers and infants.

	*n* (%)
**>Mothers (*n* = 260)**	
**Deficient (≤0.70 μmol/L)**	27 (10.4)
**Insufficient (>0.70–1.05 μmol/L)**	100 (38.4)
**Adequate (>1.05 μmol/L)**	133 (51.2)
*** Infants (*n* = 260)**	
**Deficient (≤0.70 μmol/L)**	195 (75.0)
**Insufficient (>0.70–1.05 μmol/L)**	61 (23.5)
**Adequate (>1.05 μmol/L)**	4 (1.5)

* Retinol status based on criteria by the World Health Organization for adult populations.

**Table 4 biomedicines-08-00321-t004:** Proportion of infant retinol status by maternal retinol category.

Maternal Retinol Status	Infant Retinol Status		
	Deficient	Insufficient	Adequate
	*n (%)*	*n (%)*	*n (%)*
**Deficient (*n* = 27)**	19 (70.4)	7 (25.9)	1 (3.7)
**Insufficient (*n* = 100)**	74 (74.0)	25 (25.0)	1 (1.0)
**Adequate (*n* = 133)**	102 (76.7)	29 (21.8)	2 (1.5)

**Table 5 biomedicines-08-00321-t005:** Comparison of retinol concentrations and intrauterine transfer rate between maternal retinol categories.

	Deficient (*n* = 27)	Insufficient (*n* = 100)	Adequate (*n* = 133)	Deficient vs. Insufficient (*p*-Value)	Insufficient vs. Adequate (*p*-Value)	Deficient vs. Adequate (*p*-Value)	All Groups (*p*-Value)
**Placenta ^a^ Median**	0.10	0.11	0.13	0.14	0.13	0.003	0.012
**Range (IQR)**	0.05–0.14 (0.03)	0.04–0.25 (0.06)	0.06–0.82 (0.04)				
**Maternal Serum ^b^**	0.64	0.93	1.27	<0.0001	<0.0001	<0.0001	<0.0001
	0.30–0.70	0.71–1.05	1.06–2.13				
	(0.15)	(0.16)	(0.30)				
**Umbilical Cord Blood ^b^**	0.61	0.55	0.57	0.19	0.29	0.50	0.35
	0.34–1.29	0.18–1.17	0.05–1.21				
	(0.26)	(0.27)	(0.24)				
**Intrauterine Transfer Rate ^c^**	103.4%	61.2%	43.1%	<0.0001	<0.0001	<0.0001	<0.0001
	53.1–430.0	19.1–130.6	4.3–100.8				
	(58.5)	(26.7)	(17.8)				

^a ^μg/g; ^b^ μmol/L; ^c^ calculated for each mother–infant dyad as ((umbilical cord blood retinol concentration μg/L)/(maternal retinol concentration μg/L)) × 100; *p*-values reported from Mann–Whitney U and Kruskal–Wallis testing. Bold type: reported medians.

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
