# Peer review of "Effect of Maternal Retinol Status at Time of Term Delivery on Retinol Placental Concentration, Intrauterine Transfer Rate, and Newborn Retinol Status"

_biomedicines, 2020, doi:10.3390/biomedicines8090321_

Round 1

Reviewer 1 Report

Congratulations to the authors upon writing a first-class manuscript.  I am pleased to state that I have very few scientific nor stylistic concerns.

My concerns are identified below:

l. 72: I see that the subjects gave informed consent for inclusion; however, I presume that they also gave permission for their newborns to participate. Please include this in your description.  If you did not get permission, this would constitute a serious breach of ethics.

l. 147: Table 3. for gestational age, shouldn't the data for Median and Maximum, state + or-? Also state whether this is an estimate or a statistical calculation, and if so, whether these are SD's or SE's.

l. 149, Table 2, under race, substitute "Indigenous Americans" for American Indians as that would be more accurate.

l. 248: Indubitably, this comment regarding "maternal deficiency during pregnancy" is extremely important and worthy of a further study.

l. 268-273: No question that women need to focus on provitamin A intake as do the newborns. I ran a study on beta-carotene supplementation to pregnant women and the level of beta-carotene in their breast milk (unpublished results), but this highlights the need for including provitamin A, at least beta-carotene intake and its measurement, in subsequent studies.    

l. 280: Change "a" to "at".

Author Response

Thank you again for such valuable feedback.  Please see the attachment.

Reviewer 2 Report

This manuscript presents "retinol placental concentration, intrauterine transfer, and neonatal status at time of term delivery between cases of maternal retinol adequacy, insufficiency, and deficiency in a United States population". 

The manuscript is well presented. I have some overall comments:

Methods:

1) Was there any correlation between retinol concentrations and birth weight?

Results:

2) Figure 1 needs to be presented more clearly. Please provide y-axis title. This graph was not reproduced clearly in the version of the manuscript for review, as part of it seems to be missing.

3) The authors define "intrauterine transfer rate" as (umbilical cord blood retinol conc/ maternal retinol conc) X 100. Please clarify if it is supposed to be transplacental transfer rate or intrauterine transfer rate. This calculation seems to represent more of a ratio of fetal to maternal concentration at a specific snapshot in time and not a rate. The authors also mention that retinol was not measured at the start of pregnancy (or as part of a time course). 

Discussion:

4) The authors mention (line 185) "...raw median levels of retinol in umbilical cord blood were highest among infants born to retinol deficient compared to insufficient or adequate mothers". Please discuss.

5) Please provide discussion of mechanisms of transfer of retinol and its potential effects on the placenta (deficient vs insufficient vs adequate) and therefore the fetus during development. 

Author Response

Thank you for reviewing this manuscript and providing valuable feedback.  Please see the attachment.
